# Study of the Heat Transfer Performance of Laminated Paper Honeycomb Panels

**DOI:** 10.3390/biomimetics8010046

**Published:** 2023-01-19

**Authors:** Yinsheng Li, Jing Yang, Jinxiang Chen, Jian Yin

**Affiliations:** 1Key Laboratory of Concrete and Prestressed Concrete Structures of the Ministry of Education, Southeast University, Nanjing 211189, China; 2Suzhou Bo Guan Yue Qu Intelligent Equipment Co., Ltd., Suzhou 215000, China

**Keywords:** laminated paper honeycomb panels, heat transfer model, radiation heat transfer, structural parameters

## Abstract

To apply functional honeycomb panels (FHPs) in actual engineering projects, the heat transfer performance and intrinsic heat transfer mechanism of laminated honeycomb panels (LHPs, total thickness of 60 mm) with different structural parameters were investigated in this study by a heat flow meter. The results showed that (1) the equivalent thermal conductivity *λ*_equ_ of the LHP was almost independent of the cell size, when it consisted of a small single-layer thickness. Thus, LHP panels with a single-layer thickness of 15–20 mm are recommended. (2) A heat transfer model of LHPs was developed, and it was concluded that the heat transfer performance of LHPs depends greatly on the performance of their honeycomb core. Then, an equation was derived for the steady state temperature distribution of the honeycomb core. (3) The contribution of each heat transfer method to the total heat flux of the LHP was calculated using the theoretical equation. According to the theoretical results, the intrinsic heat transfer mechanism affecting the heat transfer performance of LHPs was revealed. The results of this study laid the foundation for the application of LHPs in building envelopes.

## 1. Introduction

As early as the last century, honeycomb sandwich structures were used in areas such as aerospace, building insulation, and solar collectors [1,2]. Theoretical studies on the heat transfer of honeycomb panels (HPs) are also making progress. In the last century, Swann et al. [3] proposed a semi-empirical formula for calculating the equivalent thermal conductivity of honeycomb sandwich structures; the formula mainly considered the influence of material properties and structural dimensions on the results. Arulanantham et al. [4,5] proposed a heat transfer model for square honeycomb insulation with coupled radiation and conduction and discussed the key influencing factors. Daryabeigi [6] further modified the Swann–Pittman semi-empirical relationship by adding face and bond layers to the thermal model to give a heat transfer control equation with coupled radiation and conduction in a hexagonal honeycomb and obtained more accurate calculation results. On this basis, assuming that the temperature is distributed quadratically along the thickness of the outer panel and linearly along the thickness of the honeycomb core, Zhang et al. [7] proposed a temperature shell element with three layers and a total of 32 nodes that can consider multiple boundary conditions. It improved the efficiency of the computation of the transient temperature field of the honeycomb sandwich panel. Tang et al. [8] proposed a method to approximate radiation heat transfer inside a cell with a converted interlayer radiation coefficient. Then, the macroscopic equivalent thermal conductivity of the three-layer HP was obtained by analyzing the radiation boundary conditions and the correlation between the temperature gradient and thermal performance. Hou et al. [9] used a transfer matrix method to calculate the total heat transfer of a multi-cavity honeycomb; the results showed that the out-of-plane heat transfer performance of a multi-cavity honeycomb is better than that of a conventional honeycomb.

Meanwhile, to improve the thermal insulation performance of honeycomb sandwich panels, new structures have emerged. For example, Hum et al. [10] developed a composite honeycomb structure with an air gap that resulted in a 39% reduction in total thermal conductivity. Shi et al. [11] explored the energy band structure of a sandwich panel with cruciform trabeculae and calculated its equivalent thermal resistance. Sun et al. [12] properly solved the thermal control problem between a flexible thin-film solar cell and a blimp shell by using a Nomex honeycomb based on the preparation of a new multilayer insulation material. Additionally, new insulation materials were applied to honeycomb structures. For example, Berkefeld et al. [13] investigated the heat transfer performance of honeycomb composites made of paper honeycomb as the skeleton and aerogel as the filling material. Wang et al. [14] prepared a composite honeycomb core layer with SiO_2_ powder and SiC reticulated foam to effectively suppress thermal radiation. Most of the above studies focused on conduction and radiation by honeycomb sandwich panels at high temperatures, while there were few studies at room temperature.

However, more than 20 years ago, our group studied the three-dimensional structure of the elytron of T. dichotomus and Lucanidae [15] and were inspired to propose a honeycomb sandwich panel with trabecular-honeycomb structure, the beetle elytron panel, or BEP (Figure 1) [16,17]. In recent years, we have cooperated with manufacturers [18] to jointly develop new products and construction technologies derived from hollow floors, such as BEPs for wall insulation. It has been proven that BEPs have better mechanical properties than HPs in terms of compression [19], bending [20], and shear [21]. In previous papers [22,23,24,25], the thermal insulation performance of BEPs was also investigated. In view of the difficulty and cost of BEP preparation, the HP was regarded as the simplest BEP (the column diameter in Figure 1d is 0), and it was taken as the research object in the study for the heat transfer performance of BEPs. With this assumption, the three basic heat transfer methods were considered and calculated theoretically, and the mechanism for the influence of the honeycomb size, plate thickness, and filling material on the heat transfer performance of a single-layer HP were investigated. It was demonstrated that the results of HP heat transfer studies are fully applicable to BEPs, and their conclusions are more conservative than those derived from BEPs. On this basis, to utilize its thermal insulation performance in the building field, this paper takes multi-layer HPs as the research object and further investigates the heat transfer mechanism of the laminated honeycomb panel (LHP) to lay the foundation for developing lightweight thermal insulation building materials such as functional beetle elytron panels (FBEPs) and functional honeycomb panels (FHPs) that also possess excellent mechanical properties. Therefore, this paper develops a theoretical model of heat transfer of LHPs and analyzes the heat transfer mechanism of LHPs with cell sizes and panel thicknesses used commonly in the construction field. The model gives controlling equations for the temperature distribution inside the LHP. The heat transfer performance and mechanism of LHPs with different single-layer honeycomb thicknesses and honeycomb edge lengths are investigated for the same total thickness.

## 2. Materials and Methodology

### 2.1. Sample Design and Experimental Materials

Considering that the common thickness of the precast concrete hollow panel is 120 mm, excluding the thickness of the upper and lower panels and the waterproof layer outside the paper honeycomb, the thickness of the LHP is taken as 60 mm. The design scheme of LHPs with the same plate thickness (h = 60 mm) and different core structure parameters is given in Figure 2.

The paper used for the honeycomb core and panel in this study was A-grade Kraft paper produced by Jurong Dongshan Paper Factory. Its properties were as follows: (1) The honeycomb core paper was available in two different sizes, 16 mm and 8 mm edge lengths (Figure 2a,b); the measured density was 687.50 kg/m^3^, the thermal conductivity *λ* was 0.09 W/(m·K), the paper thickness was 0.16 mm, and the available single-layer thicknesses were 10, 15, 20, 30, and 60 mm. (2) For the panel paper, the measured density was 850.00 kg/m^3,^ and the paper thickness was 0.20 mm. The binder is UFO glue (executive standard: HG/T 5377-2018) produced by Yiwu Sky Adhesive Products Co., Ltd (Yiwu, Zhejiang, China)., and its main components were polymer and water-soluble resin.

### 2.2. Preparation of LHPs

The preparation of an LHP, shown in Figure 3, is carried out as follows: (1) Stretch and shape the honeycomb by unfolding the compressed honeycomb paper and stretching it to the expected width. Fix the ends of the honeycomb with heavy blocks and ensure that each unit is properly extended with a shape similar to a hexagon, adjusting as necessary. Keep the heavy blocks in place for more than 24 h to fully set the stretched honeycomb. (2) Bond, compact, and cure a single-layer HP with one panel. To ensure full coherence between the honeycomb and panel, apply glue uniformly on the panel. To prevent the honeycomb from bulging due to uneven shrinkage of the glue, cure the bonded honeycomb board at a pressure of 500 Pa and room temperature for 24 h. (3) Cut and trim the HP by cutting off both ends compacted by the heavy blocks. Trim the HP to a standard dimension of 300 × 300 mm with the help of a stainless-steel panel of the same size. (4) Bond the single-layer HPs together. Use the panels adjacent to two HPs as a communal surface and use glue to bond the stacked HPs on the communal panels. (5) Cap the LHP by preparing a separate 300 × 300 mm panel and gluing it to the top of the LHP by the same method to complete the seal. (6) Compact and cure the LHP by trimming, shaping, and then accelerating the solidification of the binder at 500 Pa and 40 °C. After curing for 48 h, store the LHPs at room temperature.

### 2.3. Experimental Methods and Data Acquisition

Thermal conductivity measurements based on the heat flow method were conducted for the specimens (Figure 4b). The experimental apparatus was a heat flow meter (HFM 436/3/1E) with a repeatability of 0.25% and accuracy of ± 1–3%. The hot and cold plate temperatures were set to 40 °C and 25 °C, respectively. The values were determined considering the physical comfort and summer indoor average temperature in the central and southern regions with higher populations in China, as well as to better compare with previous data [22,23]. The sample was sandwiched between the two heat flow sensors. By controlling the elevation of the upper plate, the specimen and the heating plate were in close contact over the entire surface to obtain the smallest and most uniform contact thermal resistance, and to ensure the repeatability of the thermal conductivity measurements. All test parameters and calibration data for each specimen in the test were automatically recorded by the control system; some of the specimens are shown in Figure 4a.

Notably, the standard dimensions of the specimens were 300 × 300 mm, while the working plane of the HFM 436/3/1E was 305 × 305 mm. Thus, there was indeed a small gap on the surrounding sides of the LHP during the measurement process. However, considering that when the LHP was placed in the center, the air thickness *δ* on both sides was small (*δ* = 2.5 mm), and air was maintained under the upper hot and lower cold conditions, the possible impact of those air gaps on the measurement results was ignored.

## 3. Results and Discussion

The graphs of the equivalent thermal conductivity *λ*_equ_ for the LHP_8_ (honeycomb edge length of 8 mm) and the LHP_16_ (honeycomb edge length of 16 mm) specimens with various single-layer thicknesses and the same total thickness (*h* = 60 mm) in the upper hot and lower cold conditions are presented in Figure 5. As shown in Figure 5, with increasing single-layer honeycomb thickness, i.e., with decreasing number of stacked layers, the equivalent thermal conductivity of both the LHP_8_ and LHP_16_ specimens significantly increased. The growth rate of *λ*_LHP16_ was significantly faster, and the value of *λ*_LHP16_ was always larger than that of *λ*_LHP8_. The specific results showed that the trend could be divided into two stages. When the single-layer honeycomb thickness was small (*h*_ci_ = 10 mm, 15 mm), *λ*_LHP8_ and *λ*_LHP16_ were similar, while with a further increase in the single-layer honeycomb thickness (*h*_ci_ = 20, 30, and 60 mm), the difference between *λ*_LHP8_ and *λ*_LHP16_ increased gradually. Thus, considering the complex working conditions that paper honeycomb panels will meet when they are applied to building envelopes as insulation panels and the convenience of filling the cavities with other insulation materials, LHP panels with a single-layer honeycomb thickness of 15–20 mm are recommended to obtain a more stable heat transfer coefficient and excellent insulation effect.

### 3.1. Characteristics of LHP Heat Transfer Performance

In this section, based on the characteristics of LHP heat transfer performance, a theoretical formula of the LHP equivalent thermal conductivity and heat transfer model are established. The mechanism of the influence of structural parameters on the heat transfer performance of LHPs is explored by theoretical derivation and analysis of experimental data.

To further promote the application of LHP panels in building envelopes, we need to understand not only the various characteristics of LHP heat transfer performance and the influence by factors such as structural parameters and heat transfer direction but also the mechanisms of these influences. Thus, in the following part of this section, a heat transfer model of LHP is derived from theoretical calculations, and a qualitative analysis of the heat transfer mechanism of LHP is completed. This lays the foundation for exploring the mechanism of the influence of structural parameters and heat transfer direction on the heat transfer performance of LHPs.

### 3.2. Theoretical Model and Derivation of an Equivalent Thermal Conductivity Equation for LHPs

#### 3.2.1. Equivalent Thermal Conductivity of LHP

The LHP can be regarded as an array of adjacent regular hexagonal honeycomb cells in which simultaneous heat transfer occurs (Figure 6a). When the heat transfer reaches the steady state, there is no heat transfer between the individual components in the direction of the vertical temperature difference. Then, a cell can be extracted from the LHP heat transfer model as a basic unit, as shown in Figure 6b. The heat transfer path inside the basic unit can be depicted as follows: upper panel → honeycomb core (including honeycomb wall and air) →manual panel →...... →manual panel → honeycomb core (including honeycomb wall and air) → lower panel. The equivalent thermal conductivity of the LHP *λ*_equ_ can be calculated after separately identifying the thermal conductivity of each layer.

As shown in Figure 6c, the basic unit of the LHP heat transfer model with *n* layers can be split into *n* + 1 panels and *n* honeycomb cores, and the LHP *λ*_equ_ value can be calculated according to the following equation based on the series principle of thermal resistance,
(1)$$R_{equ}=\overset{n+1}{\underset{i=1}{\sum }}R_{fi}+\overset{n}{\underset{i=1}{\sum }}R_{ci}$$
(2)$$R_{fi}=\frac{h_{fi}}{\lambda_{fi}A_{fi}},\;R_{ci}=\frac{h_{ci}}{\lambda_{ci}A_{ci}}$$
(3)$$\lambda_{equ}=\frac{h}{R_{equ}A}$$
(4)$$h_{LHP}=\overset{n+1}{\underset{i=1}{\sum }}h_{fi}+\overset{n}{\underset{i=1}{\sum }}h_{ci}$$
where *R*_equ_ and *λ*_equ_ are the equivalent thermal resistance and equivalent thermal conductivity of the whole unit, *R*_fi_ and *λ*_fi_ are the equivalent thermal resistance and equivalent thermal conductivity of the panel in the *i*-th layer, *R*_ci_ and *λ*_ci_ are the equivalent thermal resistance and equivalent thermal conductivity of the honeycomb core in the *i*-th layer, *A* and *h* are the total area and thickness of the unit, and *n* is the number of stacked layers. Since *A* = *A*_fi_ = *A*_ci_, the above equation can be simplified to:(5)$$\lambda_{equ}=\frac{h}{\left(\sum_{i=1}^{n+1}\frac{h_{fi}}{\lambda_{fi}}+\sum_{i=1}^{n}\frac{h_{ci}}{\lambda_{ci}}\right)}$$

In this experimental condition, as *h*_fi_ = 0.2 mm ≪ *h*_ci_ and *λ*_fi_ = 0.09 W/(m·K), which is expected to have a similar order of magnitude to *λ*_ci_, the term *h*_fi_/*λ*_fi_ in the denominator can be omitted. Then, the formula can be simplified as follows:(6)$$\lambda_{E}=\frac{h_{LHP}}{\sum_{i=1}^{n}\frac{h_{ci}}{\lambda_{ci}}}$$

Obviously, the key to solving for the equivalent thermal conductivity of the LHP *λ*_equ_ is to identify the equivalent thermal conductivity of the honeycomb core *λ*_ci_.

#### 3.2.2. Heat Transfer Model and Equivalent Thermal Conductivity of Honeycomb Cores

For the honeycomb core heat transfer model whose upper and lower panels are at different constant temperatures, heat is transferred by conduction through the honeycomb walls and panels, convection of air inside the honeycomb driven by the temperature difference, and radiation between the inner surfaces of the honeycomb core. In this case, the comprehensive heat transfer includes heat conduction, convection and radiation, which are complex to consider and calculate at the same time. Therefore, approaching the heat transfer from the point of view of the heat transfer method, the total heat flow can be decomposed as follows: (1) conduction heat flow *Q*_s_ through the solid part of the honeycomb wall; (2) radiation heat flow *Q*_f_ between the upper and lower panels and the inner surfaces of the honeycomb cells; and (3) convection heat flow *Q*_a_ of the air trapped inside the honeycomb core.
(7)$$Q=Q_{s}+Q_{f}+Q_{a}$$

For the honeycomb cell shown in Figure 7a, the honeycomb walls can be approximated as having no temperature difference in the direction of thickness *t* because the walls are made of paper that is very thin. Therefore, it can be assumed that the temperature of the honeycomb wall is distributed along the height in one dimension, i.e., it is only related to the coordinates of the honeycomb wall in the height direction *h*. Therefore, the honeycomb wall can be divided into *n* uniform cells along the height direction, as shown in Figure 7b, and each of them is assumed to be an isothermal cell. Since the air convection is small under the upper hot and lower cold conditions, it can be considered that there is approximately no transverse heat transfer between the honeycomb walls and the entrapped air, and the heat transfer through the air and honeycomb walls are independent of each other. At the same time, as the air heat transfer part is relatively small and its coupling effect with the conduction heat transfer and radiation heat transfer is ignored, the heat transfer through the honeycomb wall by heat conduction and heat radiation can be considered independently.

First, for the conduction heat transfer of cell *m*, the conduction heat transfer Δ*Q*_s−m,m−1_ from cell m^−1^ to cell m can be derived by applying the finite difference similar to the previous article [26]:(8)$$\Delta Q_{s-m,m-1}=A_{s}\lambda_{s}\frac{T_{m-1}-T_{m}}{l_{m,m-1}}\Delta t$$
where *A*_S_ is the cross-sectional area of the solid heat conduction path, *λ*_s_ is the heat transfer coefficient of the honeycomb material, *l*_m,m−1_ is the distance from the center of cell m−1 to cell m, *T*_m−1_ is the temperature of cell m−1, *T*_m_ is the temperature of cell m, and $\Delta_{t}$ is the time interval. Similarly, the heat transfer from cell m + 1 to cell m by solid heat transfer Δ*Q*_s−m,m + 1_ can be obtained:(9)$$\Delta Q_{s-m,m+1}=A_{s}\lambda_{s}\frac{T_{m+1}-T_{m}}{l_{m,m+1}}\Delta t$$

Second, for cell m, the net radiative heat flow Δ*Q*_f−m_ is the sum of the radiative heat flow between it and the other surfaces of the closed cavity:(10)$$\Delta Q_{f-m}=\sigma \left[\overset{n}{\underset{k=1}{\sum }}A_{k}\varepsilon_{k}\left(G_{k,m}-\delta_{k,m}\right)T_{k}^{4}\right]\Delta t$$
where *σ* is the Stefan–Boltzmann coefficient, *A_k_* is the radiation surface area, *ε* is the emissivity, *δ* is the Kronecker symbol, and $G_{k,m}$ is the net radiation coefficient, which implies the percentage of energy radiated from surface k in the closed cavity that is absorbed by surface m after infinite reflections. The value of *G* is taken as a function of the view factor *F* and absorptance *α*, which is shown in Equation (11). The view factor of the honeycomb core can be calculated according to the process in Ref. [3]. The net radiation coefficient *G* is related only to the geometric configuration and emissivity of the honeycomb cells, and each LHP included in this paper consists of a stack of identical honeycomb cores. Therefore, for each configuration, the *G*-matrix is computed for only one core and called repeatedly when deriving the control equation for the LHP temperature distribution,
(11)$$\left[G\right]={\left[\left[1\right]-\left[F\right]\left[E\right]\right]}^{-1}\left[F\right]\left[\varepsilon \right]$$
where [*G*] is the net radiation coefficient, [*F*] is the view factor, [*ε*] is the radiation absorptivity (assuming the paper honeycomb is a gray body whose absorptivity is equal to the emissivity), and [*E*] is the radiation reflectance.

Thus, the total heat transfer of solid conduction and radiation for cell m is the sum of Equations (8)–(10).
(12)$$\begin{array}{c} \Delta Q_{m}=\Delta Q_{s-m}+\Delta Q_{f-m}=\Delta Q_{s-m,m-1}+\Delta Q_{s-m,m+1}+\Delta Q_{f-m}=\\ \left(A_{s}\lambda_{s}\frac{T_{m-1}-T_{m}}{l_{m,m-1}}+A_{s}\lambda_{s}\frac{T_{m+1}-T_{m}}{l_{m,m+1}}+\sigma \overset{n}{\underset{k=1}{\sum }}A_{k}\varepsilon_{k}\left(G_{k,m}-\delta_{k,m}\right)T_{k}^{4}\right)\Delta t \end{array}$$

The transfer of heat eventually causes the temperature of cell *m* to change, which in turn gives:(13)$$\Delta Q_{m}=m_{m}c_{p,m}\Delta T_{m}$$
where *m*_m_, *c*_p,m_, and Δ*T*_m_ are the mass, mass specific heat capacity, and temperature change of cell *m*, respectively.

Combining Equations (12) and (13) and taking the limit Δ*t*→0, the equation for the instantaneous heat transfer temperature distribution of the honeycomb core can be obtained as:(14)$$m_{m}c_{p,m}\frac{dT_{m}}{dt}=A_{s}\lambda_{s}\frac{T_{m-1}-T_{m}}{l_{m,m-1}}+A_{s}\lambda_{s}\frac{T_{m+1}-T_{m}}{l_{m,m+1}}+\sigma \overset{n}{\underset{k=1}{\sum }}A_{k}\varepsilon_{k}\left(G_{k,m}-\delta_{k,m}\right)T_{k}^{4}$$

Since the problem studied in this paper is the steady-state heat transfer performance of the honeycomb core, it is given that $\frac{dT_{m}}{dt}=0$. When it is substituted into Equation (14), the controlling equation of the temperature field distribution of the honeycomb core in the steady state is obtained.
(15)$$A_{s}\lambda_{s}\frac{T_{m-1}-T_{m}}{l_{m,m-1}}+A_{s}\lambda_{s}\frac{T_{m+1}-T_{m}}{l_{m,m+1}}+\sigma \overset{n}{\underset{k=1}{\sum }}A_{k}\varepsilon_{k}\left(G_{k,m}-\delta_{k,m}\right)T_{k}^{4}=0$$

The above formulas are applied to the model of the LHP. Let an LHP have *n* layers of honeycomb core ($h_{\mathrm{LHP}}=nh_{\mathrm{ci}}$), and each layer is divided into 10 cells in the thickness direction. Then, the thickness of each cell is $h_{ci}/10$. Since the panels are involved in radiative heat transfer, panels and cells are numbered from top to bottom as cell 1 to cell 11 *n* + 1 in sequence, and the temperatures of cell 1 and cell 11 *n* + 1 are known and kept constant. Therefore, the temperature distribution control equations for each cell are columned except cell 1 and cell 11 *n* + 1. Figure 7 illustrates the distance *l* between the centers of adjacent cells.
(16)$$\begin{array}{c} l_{11k+1,11k+2}=\frac{h_{ci}}{20}=\frac{h_{LHP}}{20n},\;l_{11k+s-1,11k+s}=\frac{h_{ci}}{10}=\frac{h_{LHP}}{10n},\;l_{11\left(k+1\right),11\left(k+1\right)+1}=\frac{h_{ci}}{20}=\frac{h_{LHP}}{20n},\\ k=0,1,2,3\ldots n-1,s=3,4\ldots 11 \end{array}$$

Substituting the detailed information of the LHPs into Equation (15), the controlling equations for the temperature field distribution of the study models can be obtained:

Single-layer LHP, *n* = 1:
(17)$$\left\{ \begin{array}{l} 10\frac{\lambda_{s}A_{s}}{h}\left(2T_{m-1}-3T_{m}+T_{m+1}\right)+\sigma \sum_{k=1}^{12}A_{k}\varepsilon_{k}\left(G_{k,m}-\delta_{k,m}\right)T_{k}^{4}=0,m=2\\ 10\frac{\lambda_{s}A_{s}}{h}\left(T_{m-1}-2T_{m}+T_{m+1}\right)+\sigma \sum_{k=1}^{12}A_{k}\varepsilon_{k}\left(G_{k,m}-\delta_{k,m}\right)T_{k}^{4}=0,m=3\ldots 10\\ 10\frac{\lambda_{s}A_{s}}{h}\left(T_{m-1}-3T_{m}+2T_{m+1}\right)+\sigma \sum_{k=1}^{12}A_{k}\varepsilon_{k}\left(G_{k,m}-\delta_{k,m}\right)T_{k}^{4}=0,m=11 \end{array}\right.$$


Multilayer LHP, *n* = 2,3,4,5,6
(18)$$\left\{ \begin{array}{c} 10\frac{\lambda_{s}A_{s}}{h}\left(2T_{m-1}-3T_{m}+T_{m+1}\right)+\sigma \sum_{k=1}^{12}A_{k}\varepsilon_{k}\left(G_{k,m-11k}-\delta_{k,m-11k}\right)T_{k}^{4}=0,m=11l+2\\ 10\frac{\lambda_{s}A_{s}}{h}\left(T_{m-1}-2T_{m}+T_{m+1}\right)+\sigma \sum_{k=1}^{12}A_{k}\varepsilon_{k}\left(G_{k,m-11k}-\delta_{k,m-11k}\right)T_{k}^{4}=0,m=11l+s-1\\ 20\frac{\lambda_{s}A_{s}}{h}\left(T_{m-1}-2T_{m}+T_{m+1}\right)+\sigma \sum_{k=1}^{12}A_{k}\varepsilon_{k}\left(G_{k,m-11k}-\delta_{k,m-11k}\right)T_{k}^{4}+\\ \sigma \sum_{k=1}^{12}A_{k}\varepsilon_{k}\left(G_{k,m-11(k+1)}-\delta_{k,m-11(k+1)}\right)T_{k}^{4}=0,m=11l+12\\ 10\frac{\lambda_{s}A_{s}}{h}\left(T_{m-1}-3T_{m}+2T_{m+1}\right)+\sigma \sum_{k=1}^{12}A_{k}\varepsilon_{k}\left(G_{k,m}-\delta_{k,m}\right)T_{k}^{4}=0,m=11l+11 \end{array}\right.$$
where l=0,1,2,3…n−1,s=3,4…11Thus far, if the temperatures of the top and bottom panels *T*_f1_ and *T*_f11n + 1_, the height of LHP *h*, the honeycomb edge length, and the honeycomb wall thickness *t* are known, then the temperature distribution of the honeycomb core can be solved by Equations (17) or (18).

Based on the above derivation, the heat flow density of the cold surface can be further calculated to finally obtain the equivalent thermal conductivity of the honeycomb core. To maintain the constant temperature of the upper and lower surfaces, the heat flow into the cold surface must be equal to the heat flow out of the hot surface. This gives the heat flow of radiation and solid conduction that transfer from the intermediate core layer and the hot surface to the cold surface:(19)$$\begin{array}{c} A_{11n+1}q_{e}=A_{11n+1}\left(q_{s}+q_{f}\right)\\ \;\;\;\;\;\;=20n\frac{\lambda_{s}A_{s}}{h}\left(T_{11n}-T_{11n+1}\right)+\sigma \overset{12}{\underset{k=1}{\sum }}A_{k}\varepsilon_{k}\left(G_{k,12}-\delta_{k,12}\right)T_{k}^{4} \end{array}$$

The above equations do not consider the effect of the gas entrapped in the honeycomb core. Assuming that the heat flux from the air inside the core is *q*_a_, the equivalent thermal conductivity of the honeycomb core λ_c_ can be obtained as:(20)$$\lambda_{ci}=\frac{qh_{ci}}{\Delta t}=\frac{\left(q_{a}+q_{e}\right)h_{ci}}{\left(T_{11i-10}-T_{11i+1}\right)}=\frac{\left(q_{a}+q_{f}+q_{s}\right)h_{ci}}{\left(T_{11i-10}-T_{11i+2}\right)}\;i=1,2\ldots .n$$

### 3.3. Calculation of the LHP Equivalent Thermal Conductivity and the Mechanism of Influence

This section investigates the mechanism of the influence of factors such as structural parameters and heat transfer direction on the heat transfer performance of LHPs. The experimental result of the LHP equivalent thermal conductivity is compared to the theoretical analysis of the heat transfer mechanism of the LHP based on Swann–Pittman’s empirical equation of radiation heat transfer within the honeycomb structure.

#### 3.3.1. Calculation of the Equivalent Thermal Conductivity of the Honeycomb Core

Based on the analysis of the heat transfer model of the honeycomb core in the previous section, the expression for the equivalent thermal conductivity of the honeycomb core can be obtained by substituting Equation (19) into Equation (20).
(21a)$$\lambda_{ci}=\frac{\left(q_{s}+q_{f}+q_{a}\right)h_{ci}}{\left(T_{hi}-T_{ci}\right)}=\lambda_{sci}+\lambda_{gci}+\lambda_{fci}$$
(21b)$$\lambda_{sci}=20\lambda_{s}\frac{A_{s}\left(T_{11i}-T_{11i+1}\right)}{A\left(T_{hi}-T_{ci}\right)}$$
(21c)$$\lambda_{fci}=\frac{\sigma h_{ci}}{A\left(T_{hi}-T_{ci}\right)}\overset{12}{\underset{k=1}{\sum }}A_{k}\varepsilon_{k}\left(G_{k,12}-\delta_{k,12}\right)T_{11i+k}^{4}$$
(21d)$$\lambda_{gci}=\lambda_{g}\frac{1-A_{s}}{A}$$
where *λ*_s_ = 0.09 W/(m·K) is the thermal conductivity of the paper honeycomb material, *λ*_g_ is the thermal conductivity of the air inside the honeycomb core, *λ*_sci,_ *λ*_gci,_ and *λ*_fci_ are the equivalent thermal conductivities of the honeycomb core in terms of solid conduction, air conduction, and radiation heat transfer, respectively, *A*_S_ is the section area of the honeycomb wall, *A* is the section area of the honeycomb unit, and *T*_Hi_ and *T*_Ci_ are the temperatures of the hotter surface and the cooler surface of the *i*th layer of the honeycomb. The radiation equivalent thermal conductivity can also be calculated by the empirical formula proposed by Swann and Pittman [3]:(22)$$\lambda_{fci}=0.664h_{ci}{\left(\alpha +0.3\right)}^{-0.69}\varepsilon^{1.63{\left(\alpha +1\right)}^{-0.89}}\sigma \left(T_{Hi}^{2}+T_{Ci}^{2}\right)\left(T_{Hi}-T_{Ci}\right)$$
where *α* = *h*_ci_/d is the ratio of the honeycomb core height to the inner tangent circle diameter *d* of the honeycomb core, *σ* = 5.67 × 10^−8^ W/(m^2^ −K^4^) [27] is the Stefan–Boltzmann constant, and *ε* = 0.90 is the emissivity of the inner surface of the honeycomb cell. Considering the influence of the panel and adhesive layer on the equivalent thermal conductivity, the equivalent thermal conductivity of the honeycomb core can be modified according to the equation in Ref. [6]. All solid intersections in this model are between the adhesive layer and the paper surface. The UFO glue is liquid before it condenses and will wet the paper surface. Therefore, the solid adhesive layer after condensation is in close contact with the paper, and the contact thermal resistance is ignored in the calculation.

The heat transfer through the air inside the honeycomb core should be discussed separately under different conditions. Air convection can be interpreted as heated air with lower density flowing upward, which leads to heat transfer between the cold and hot fluids when they are mixed through molecular movement. In this study, two working conditions were considered: upper hot and lower cold/upper cold and lower hot. In the former condition, the heated fluid in direct contact with the hot surface was located above the cooled fluid in direct contact with the cold surface, which made it difficult for the cold and hot fluids to mix and transfer heat. In the latter condition, the situation was the opposite, the heated fluid was located below the cooled fluid, and the density difference made the hot and light fluid flow upward; then, heat transfer was enhanced through fluid mixing. According to this model, it can be approximated that only conduction must be considered for heat transfer through the gas under the upper hot and lower cold conditions, while the enhancement effect brought by air convection on heat transfer under the upper cold and lower hot conditions could not be ignored. Since the analysis of air convection is complicated to calculate accurately by a theoretical formula, it will be analyzed by the finite element method in a subsequent study.

In this section, the equivalent thermal conductivity of the LHP is calculated only for the upper hot and lower cold conditions. First, the LHP temperature distribution is calculated based on Equations (17) and (18), taking the single-layer and three-layer LHPs with honeycomb side length 8 as an example. Then, the net radiation coefficients of 60 mm thick and 20 mm thick honeycomb cores are calculated and substituted into the temperature control equation. The temperature distributions of the stacked honeycomb cores are shown in Figure 8. The equivalent thermal conductivity *λ*_ci_ of each honeycomb core is then calculated by Equation (21) and modified. Finally, the overall equivalent thermal conductivity *λ*_equ_ is calculated by Equation (6), as shown in Table 1.

#### 3.3.2. Analysis of the Theoretical Results of the LHP Equivalent Thermal Conductivity

Figure 9 illustrates the theoretical results *λ*_EC_ and the experimental values *λ*_EE_ of the equivalent thermal conductivities of the LHPs with two different cell widths under the upper hot and lower cold conditions. Their variation and the percentage of their three heat transfer components with various single-layer honeycomb thicknesses *h*_ci_ are also presented. The difference between *λ*_EC_ and *λ*_EE_ shows that the values of *λ*_EE_ tended to be larger than those of *λ*_EC_, which was especially obvious in the LHP_8_ specimens. However, the specific gravity histograms for the three different heat transfer methods show that the proportion of the radiation heat transfer *λ*_fc_ for the LHP_8_ and LHP_16_ specimens rose with increasing single-layer thickness. The highest percentage exceeded 80% and appeared at LHP_16_, with a single layer thickness of 60 mm. This further manifested radiation as the dominant heat transfer method not only in a single-layer honeycomb panel HP at room temperature [21] but also in an LHP. The intrinsic causes of the above phenomenon are further analyzed as follows.

First, most of the *λ*_EC_ values were less than the *λ*_EE_ values; this was more pronounced in the LHP_8_ than the LHP_16_ specimens. This may have been influenced by the preparation process. For example, the limited fabrication accuracy made it difficult to ensure the airtightness of each honeycomb core and resulted in local convection; it was difficult to keep the honeycomb cores in a perfectly uniform arrangement, and the inexact correspondence of adjacent layers led to fluctuations in heat transfer and temperature distributions on the surfaces of the honeycomb walls; and the glue layer affected the emissivity and conduction of the upper and lower surfaces, especially in the LHP_8_, where more glue was needed.

Next, the effects of structural parameters on radiation heat transfer are examined. The effect of honeycomb edge length is the first to be investigated. Equation (22) shows that the honeycomb edge length mainly affected the theoretical result by influencing the ratio *α* between the height of the honeycomb core and the diameter of its inner tangent circle. From the derivation of the net radiation heat transfer, the honeycomb edge length mainly affected the weight of radiation heat transfer [22], which made that of LHP_16_ much greater than that of LHP_8_. The second factor was the effect of the single-layer honeycomb thickness. The reduction in the single-layer honeycomb thickness reduced the area of the internal radiating surface of a honeycomb core, resulting in a reduction in radiation heat transfer. Additionally, when the total plate thickness remained unchanged, the reduction in the single-layer honeycomb thickness decreased the temperature difference of each layer. Like *Q*_f_∝(*T*_1_^4^–*T*_12_^4^), the reduction eventually decreased the radiation heat transfer.

Thus far, this paper has established the heat transfer model of LHPs, revealing their heat transfer characteristics. The controlling equation of the temperature distribution under steady-state conditions has been derived by the finite difference method. The empirical equation has been applied to calculate *λ*_equ_ for the LHP under the upper hot and lower cold conditions. Then, the mechanism of the heat transfer performance of LHPs varying with structural parameters has been analyzed qualitatively. This study provides a theoretical foundation for the development of LHPs with good thermal insulation performance, broadens the application field of thermal insulation paper honeycomb panels, and suggests directions for future studies to improve the thermal insulation performance of LHPs.

## 4. Conclusions

In this study, the *λ*_equ_ of LHPs with different honeycomb edge lengths and single-layer plate thicknesses were measured. The heat transfer characteristics of the LHPs were revealed by theoretical derivation and calculation. The following conclusions were obtained.

(1)For the first time, comparative experiments were conducted for LHPs with different structural parameters at the same plate thickness. The standardized procedures for the preparation of specimens were described. The equivalent thermal conductivity λ_equ_ of the LHP under the experimental conditions in this paper was significantly superior to the value of λ_equ_ for LHP_8_ and LHP_16_ at small single-layer thicknesses (*h_ci_* = 10 and 15 mm) than at larger plate thicknesses, and was almost independent of the structural dimensions. It was recommended to preferably use LHP_16_ with a single-layer thickness of 15–20 mm to obtain better stability of the heat transfer coefficient and a good thermal insulation effect. It was also recommended that LHP_16_ with a board thickness of 15–20 mm be used preferentially due to its more stable heat transfer coefficient and better thermal insulation.(2)The main heat transfer path of the LHPs was given, and accordingly, a heat transfer model of LHPs with different honeycomb edge lengths and single-layer plate thicknesses was established. From this model, it was concluded that the heat transfer performance of the LHPs depended critically on the performance of the honeycomb core. Subsequently, the controlling equation of its temperature distribution under steady-state heat transfer was derived using the finite difference method.(3)Based on the theoretical equation of the honeycomb core equivalent thermal conductivity and the empirical equation of radiation heat transfer, the percentages of three different heat transfer methods, radiation, gas convection, and heat conduction, were obtained for the LHPs. Radiation heat transfer was the main heat transfer mode not only in the single-layer honeycomb panel HP at room temperature but also in the LHPs. Then, the mechanism of the influence of the structure parameters, honeycomb edge length, and single-layer plate thickness on the heat transfer characteristics of LHPs was also revealed.

The research results obtained in this study provide a theoretical foundation for improving the thermal insulation performance of paper honeycomb panels through the design of internal spatial structure parameters. They broaden the application scenario of paper honeycomb panels in building envelopes. These results also provide a reference for the future development of lightweight and high-performance multifunctional panels based on BEPs, which have a similar spatial structure and better mechanical properties than HPs.

## Figures and Tables

**Figure 1 biomimetics-08-00046-f001:**
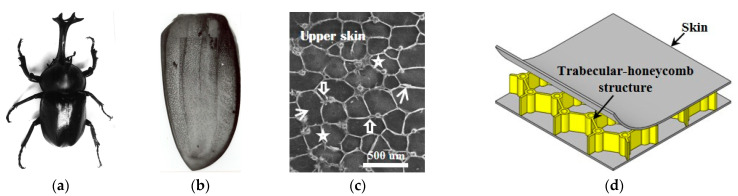
The internal structure of a beetle elytron and its biomimetic model, the BEP. (**a**) male adult T. dichotomus, (**b**) perspective view of an elytron, (**c**) internal three-dimensional structure of a beetle elytron (Big arrows are trabecula. Small arrows are honeycomb walls. Stars are honeycombs) and (**d**) structure of a BEP core layer [16,17].

**Figure 2 biomimetics-08-00046-f002:**
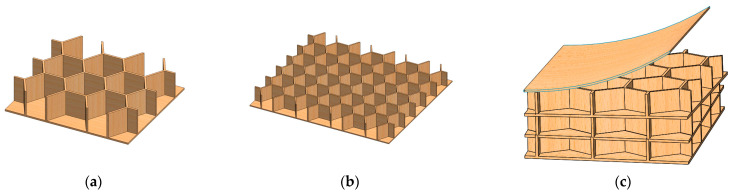
The design scheme of the three-dimensional LHP with different structural parameters. (**a**) and (**b**) internal structure of a single-layer HP with honeycomb edge lengths of 16 mm and 8 mm, respectively, and (**c**) multilayer HP: example of a three-layer LHP with single-layer height 20 mm.

**Figure 3 biomimetics-08-00046-f003:**
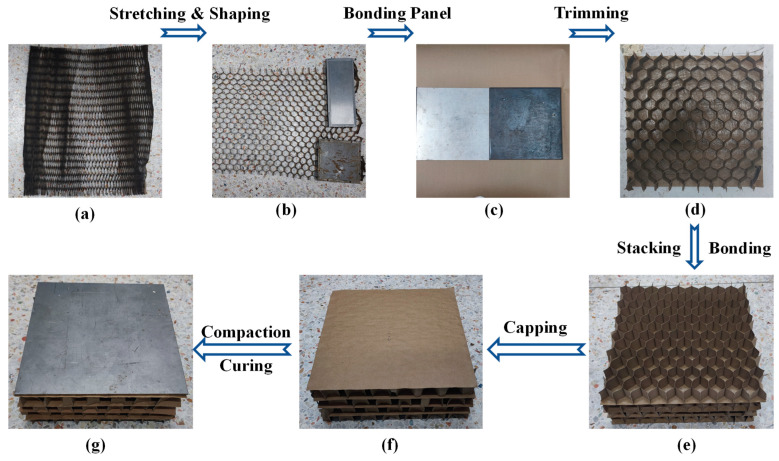
Preparation of LHPs. (**a**) unstretched honeycomb core, (**b**) stretching and shaping, (**c**) bonding a panel and a layer of honeycomb together, (**d**) cutting and trimming HPs, (**e**) stacking and sticking HPs to prepare LHPs, (**f**) capping the LHPs, and (**g**) compaction, shaping and curing of LHPs.

**Figure 4 biomimetics-08-00046-f004:**
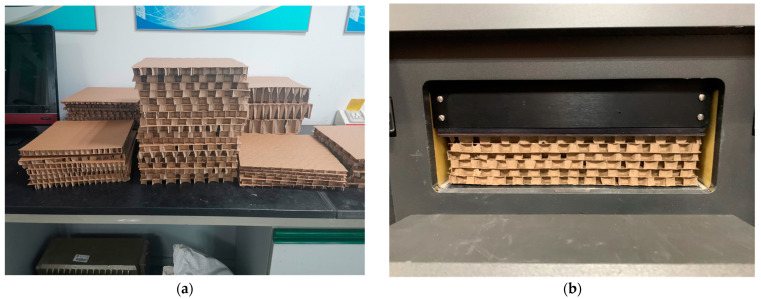
Some of the specimens and an example of their setup in the experimental test apparatus. (**a**) fabricated specimens and (**b**) equivalent thermal conductivity testing of LHPs.

**Figure 5 biomimetics-08-00046-f005:**
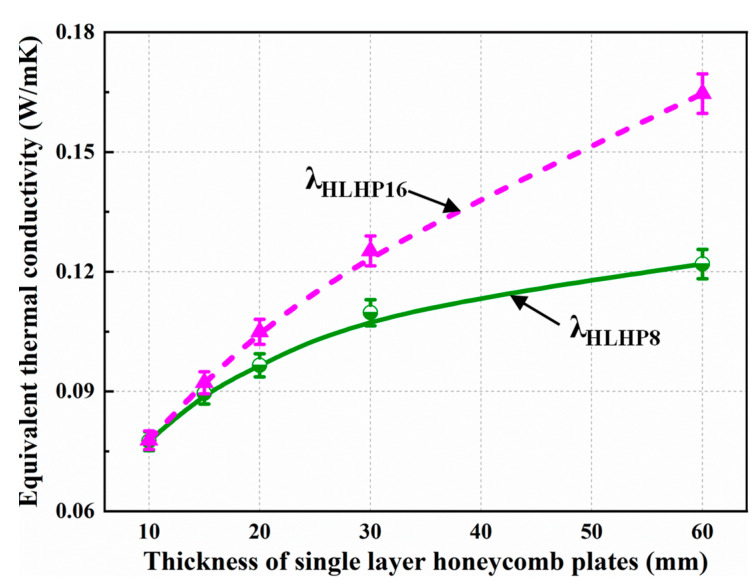
Experimental equivalent thermal conductivity for an LHP 60 mm thick with different single-layer thicknesses and honeycomb edge lengths.

**Figure 6 biomimetics-08-00046-f006:**
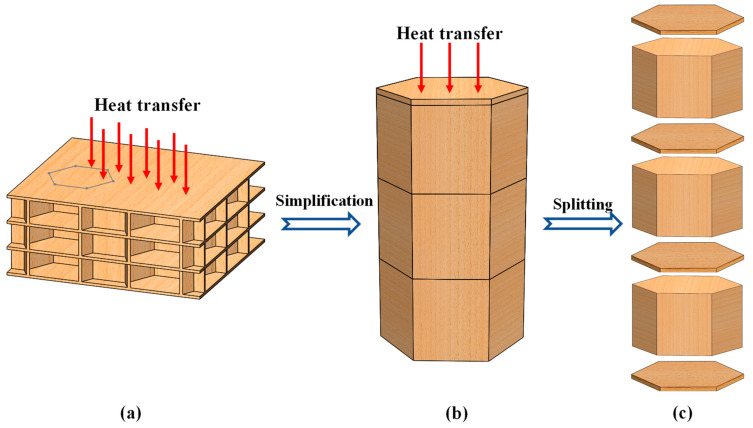
Simplified heat transfer model of the LHP (using a three-layer LHP as an example). (**a**) schematic diagram of LHP heat transfer; (**b**) LHP heat transfer model basic unit, and (**c**) separated layers of a basic unit.

**Figure 7 biomimetics-08-00046-f007:**
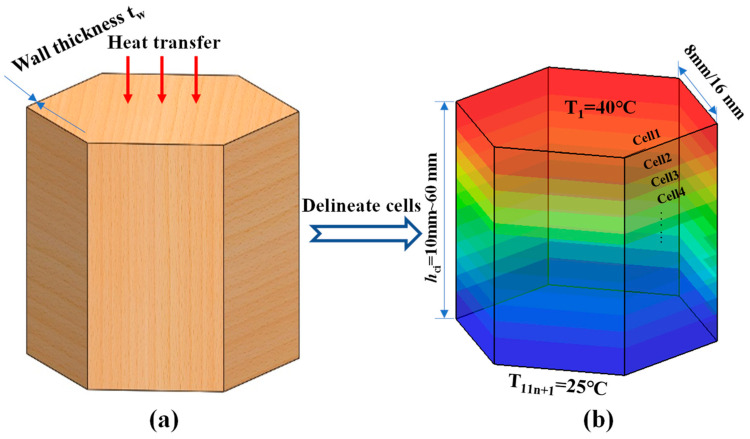
Division of an LHP honeycomb cell. (**a**) schematic diagram of a honeycomb cell and (**b**) uniform division of a honeycomb cell.

**Figure 8 biomimetics-08-00046-f008:**
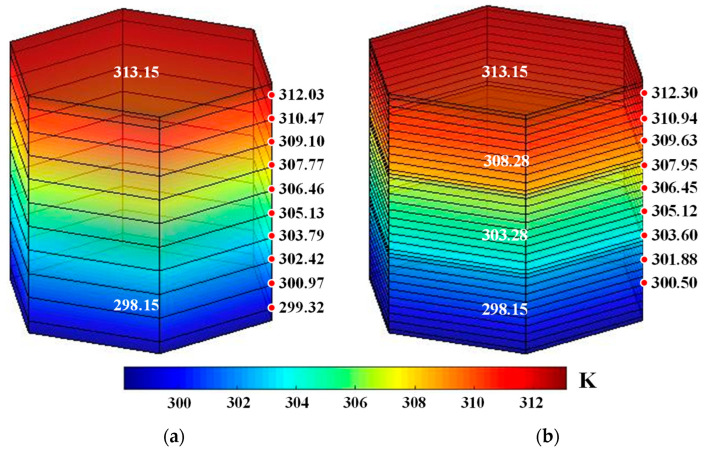
Results of theoretical calculation of the temperature distributions of single- and three-layer LHP honeycomb cells: (**a**) single-layer and (**b**) three-layer cells.

**Figure 9 biomimetics-08-00046-f009:**
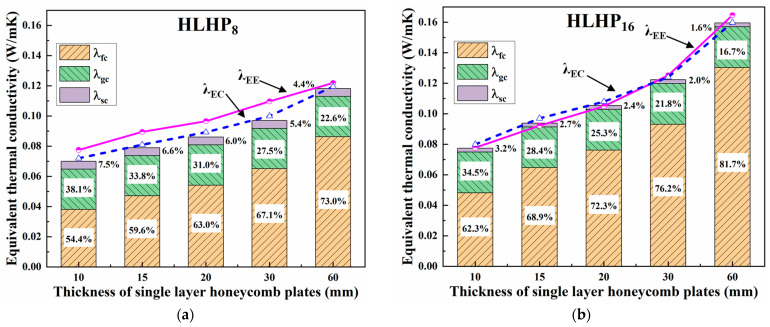
Theoretical values of the equivalent thermal conductivity of the LHP and the percentage of each heat transfer method. (**a**) LHP_8_; (**b**) LHP_16_, where *λ*_EE_ values were experimentally measured, *λ*_EC_ values were theoretical, and *λ*_fc_, *λ*_gc_, and *λ*_sc_ were the contributions of radiation, gas conduction, and solid conduction, respectively.

**Table 1 biomimetics-08-00046-t001:** Theoretical calculated values of the equivalent thermal conductivity of each honeycomb core and a monolithic plate of the LHP for upper hot and lower cold conditions.

	Honeycomb Side Length	LHP_8_				LHP_16_			
Layer Thickness		Radiation	Heat Transfer	Air	Total	Radiation	Heat Transfer	Air	Total
10	Honeycomb 1	0.0432	0.0275	0.0052	0.0759	0.0477	0.0275	0.0052	0.0804
	Honeycomb 2	0.0421	0.0272	0.0052	0.0745	0.0464	0.0272	0.0052	0.0788
	Honeycomb 3	0.0411	0.027	0.0052	0.0733	0.0450	0.027	0.0052	0.0772
	Honeycomb 4	0.0401	0.0268	0.0052	0.0721	0.0437	0.0268	0.0052	0.0757
	Honeycomb 5	0.0391	0.0266	0.0052	0.0709	0.0424	0.0266	0.0052	0.0742
	Honeycomb 6	0.0382	0.0264	0.0052	0.0698	0.0411	0.0264	0.0052	0.0727
	Integral Board	0.0406	0.0269	0.0052	0.0727	0.0443	0.0269	0.0052	0.0764
15	Honeycomb 1	0.0526	0.0274	0.0052	0.0852	0.0633	0.0274	0.0052	0.0959
	Honeycomb 2	0.0507	0.0271	0.0052	0.083	0.0607	0.0271	0.0052	0.0930
	Honeycomb 3	0.0489	0.0268	0.0052	0.0809	0.0581	0.0268	0.0052	0.0901
	Honeycomb 4	0.0471	0.0265	0.0052	0.0788	0.0556	0.0265	0.0052	0.0873
	Integral Board	0.0497	0.0269	0.0052	0.0819	0.0593	0.0269	0.0052	0.0915
20	Honeycomb 1	0.0597	0.0274	0.0052	0.0923	0.0753	0.0274	0.0052	0.1079
	Honeycomb 2	0.0568	0.0269	0.0052	0.0889	0.0712	0.0269	0.0052	0.1033
	Honeycomb 3	0.0541	0.0265	0.0052	0.0858	0.0773	0.0265	0.0052	0.0990
	Integral Board	0.0568	0.0269	0.0052	0.0889	0.0711	0.0269	0.0052	0.1033
30	Honeycomb 1	0.0701	0.0272	0.0052	0.1025	0.0933	0.0272	0.0052	0.1254
	Honeycomb 2	0.0651	0.0266	0.0052	0.0969	0.0860	0.0266	0.0052	0.1178
	Integral Board	0.0675	0.0269	0.0052	0.0996	0.0894	0.0269	0.0052	0.1315
60	Honeycomb 1/monolithic board	0.0883	0.0269	0.0052	0.1204	0.1252	0.0269	0.0052	0.1571

## Data Availability

The data that support the findings of this study are available upon reasonable request from the authors.

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
