# Peer review of "Study of the Heat Transfer Performance of Laminated Paper Honeycomb Panels"

_biomimetics, 2023, doi:10.3390/biomimetics8010046_

Round 1

Reviewer 1 Report

The paper presents findings on heat transfer performance of laminated paper honeycomb panels. In general English language needs to be improved for better readability of the script, additionally following items are suggested to work on for improving quality of the submission.

Nomenclature list needs to be added.

Reference citation needs to be corrected (Pg 1, row 31, row 41, Pg 2 row 57 etc.)

English language correction (Pg 2, row 66), and no personal nouns in scientific writing.

Methodology to be written in past tense

Pg 3, row 89 and 91 120 mm, 60 mm and so on

Units to density to be written correctly (pg 3, row 100)]

Pg 4, row 135 citation is missing

How much is the degree of uncertainty in thermal conductivity?? Pg 6 Fig 5, this is important to clarify quality of repeatability of the tests

Reviewer 2 Report

In this manuscript, the equivalent thermal conductivity λequ. of functional honeycomb panels with different honeycomb edge lengths and single-layer plate thicknesses were measured. The heat transfer characteristics of LHP were revealed by theoretical derivation and calculation. However, there are some problems must be solved before it is considered for publication. If the following problems are well-addressed, this reviewer believes that the basic contribution of this paper is important for applications such as green building and energy efficient building.

1.      In the introduction, “Most of the above studies focused on the conduction and radiation of honeycomb sandwich panels at high temperatures, while there are few studies on that at room temperature”, where the previous study was applicable under high temperature harsh conditions, will the results of the authors' study at room temperature be more generally applicable?

2.      What is the purpose of mimicking the trabecular-honeycomb structure inspired by the beetle elytron, and what are its performance advantages, please give more discussion instead of just imitating for the sake of imitation?

3.      For the preparation process of LHP, how to ensure the consistency of experimental samples?

4.      Please provide more details on the calculation and acquisition of contact thermal resistance and whether the use of adhesive between structures will result in larger contact thermal resistance.

5.         There are some spelling errors and grammar mistakes in the manuscript, the language needs to be further polished. In addition, there are some errors in the description statement of the figures (figure1d) and the masking of data by the scale bar (figure1c).
